

# Tree height-diameter allometry and implications for biomass estimates in Northeastern Amazonian forests

Aldine Luiza Pereira Baia[1], Henrique E. M. Nascimento[2], Marcelino Guedes[3], Renato Hilário[4] and José Julio Toledo[4]

[1] Post-Graduate Program in Tropical Biodiversity, Federal University of Amapá, Macapá, Amapá, Brazil
[2] Coordination of Biodiversity, National Institute for the Amazon Research, Manaus, Amazonas, Brazil
[3] EMBRAPA Amapá, Macapá, Amapá, Brazil
[4] Department of Environment and Development, Federal University of Amapá, Macapá, Amapá, Brazil

Corresponding author
José Julio Toledo, jjulio@unifap.br

## ABSTRACT

The relationship between tree height and diameter varies across forest types, introducing uncertainties in height that can affect aboveground biomass estimates in tropical forests. Here, we used a four-step approach to assess whether incorporating height estimates from local height-diameter models, compared to two published equations, improves biomass estimates across spatial scales. First, we measured the diameter and height of 1,962 trees in two representative forest types in the Northeastern Amazon: non-flooded terra-firme and seasonally-flooded várzea forests. Second, we selected the best height-diameter models from a set of 10 candidates to establish local allometric equations. Third, we applied these best local models and two previously published height models (the regional Guyana shield, and the pantropical model) to estimate tree height, and compared these estimates to measured height. Finally, we computed tree biomass using equations that both included and excluded height, and compared these biomass estimates to those calculated using directly measured height. Asymptotic height-diameter models provided the best fit at local and regional scales. The Quadratic model was the best choice for terra-firme and várzea forests separately, while the Weibull and Michaelis-Menten models performed best for both forests. Local models closely matched measured heights, with deviations of only 0.1%, outperforming the regional and pantropical models within each forest type. The regional model underestimated height in terra-firme by 3% and overestimated it in várzea by 29%, while the pantropical model underestimated height in terra-firme by 19% and overestimated it in várzea by 6%. Using local asymptotic models to estimate height improved the accuracy of biomass estimates, with differences of around 1% between biomass computed using measured heights and estimated heights for terra-firme and várzea forests. In contrast, the biomass calculated using estimated heights from both the regional and pantropical models overestimated the biomass in várzea by 41% and 17%, respectively, while the pantropical model underestimated biomass in terra-firme by 17%. The estimated height and biomass of large trees using regional and pantropical models showed the highest deviations from the observed values. Our findings underscore the necessity for height-diameter modeling for different forest

types, and highlight the need to increase sampling of large trees to improve biomass estimation accuracy in Northeastern Amazonia.

# INTRODUCTION

Uncertainties in biomass estimates for tropical forests primarily arise from substantial errors in allometric models derived from measured tree variables (*Nam, Van Kuijk & Anten, 2016*). Tree diameter is commonly used to estimate biomass indirectly, but integrating additional traits such as height, crown size, and wood density may enhance precision and accuracy (*Chave et al., 2005*; *Nogueira et al., 2008a*, *2008b*; *Chave et al., 2014*; *Goodman, Phillips & Baker, 2014*). Inter- and intra-specific variability in tree allometry and wood density (*Chave et al., 2006*; *Fayolle et al., 2016*; *Siliprandi et al., 2016*) may amplify uncertainties in allometric equations for aboveground biomass, especially in species-rich tropical forests. Moreover, the pronounced variation in tree height across climatic and edaphic gradients introduces further uncertainties into forest biomass estimates, highlighting the need to include height measurements in regular forest inventories for incorporation in allometric models (*Larjavaara & Muller-Landau, 2013*; *Fayolle et al., 2016*; *Sullivan et al., 2018*). Therefore, understanding how trait variation affects the predictability of allometric models may contribute to reducing the uncertainties in biomass estimates.

Incorporating height in biomass equations improves estimates by accounting for the intra- and inter-specific variability in traits such as height and diameter, which are influenced by site characteristics (*Nogueira et al., 2008a*; *Schietti et al., 2016*; *Fayolle et al., 2016*, *Siliprandi et al., 2016*). In Southern Amazonia, trees are shorter (*Nogueira et al., 2008a*) probably due to differences in floristic composition and climatic constraints. Soil characteristics also influence forest height by affecting forest dynamics. Tree turnover (mortality and recruitment) is higher in Western Amazonia, where soils are geologically younger, shallow, siltier, and more fertile than weathered, deeper, clay-rich soils in East-Central Amazonia and the Guyana shield (*Phillips et al., 2004*; *Quesada et al., 2012*; *Esquivel-Muelbert et al., 2020*). Soil structure and fertility gradients correlate with wood density, with Western Amazonia harboring tree species with lighter wood than the eastern portion of the basin (*Quesada et al., 2012*). At smaller scales (1–100 km) tree mortality is associated with soil structure, being higher on less stable sandy soils with shallow water tables (*Ferry et al., 2010*; *Toledo et al., 2011*, *2012*, *2017*). Different forest types also show varying turnover rates. In Amazonia, terra-firme forests show lower turnover rates than várzea forests, likely due to the deep and well-drained soils of terra-firme *vs*. the shallow and waterlogged soils of várzea (*Korning & Balslev, 1994*; *Toledo et al., 2017*). Thus, soils indirectly influence species trait selection, favoring fast-growing taxa with lighter wood, smaller, shorter trees on poorly structured, and waterlogged soils (*Quesada et al., 2012*; *Schietti et al., 2016*; *Toledo et al., 2017*). However, the height-diameter relationships were

not compared between forest types which evolved in contrasting edaphic conditions, such as terra-firme and várzea forests.

Species with distinct life histories show different architectural relationships (*e.g.*, height-diameter, crown width-height) which vary within and between sites (*Poorter, Bongers & Bongers, 2006*; *King et al., 2006*; *Iida et al., 2012*; *Siliprandi et al., 2016*). Thus, differences in species composition within and between forest types are expected to drive variation in allometric relationships, similar to how variations in species composition lead to variability in traits like diameter, height, and wood density (*Quesada et al., 2012*; *Schietti et al., 2016*). Species composition varies widely at local (*Damasco et al., 2013*; *Toledo et al., 2017*) and regional (*ter Steege et al., 2006*) scales in Amazonia and partially influences height-diameter relationships. Nonetheless, the extent to which species composition affects these relationships remains under investigation.

Competition likely plays an important role in explaining variations in height-diameter allometry between Amazonian forests. Neighborhood crowding (a competition indicator) disproportionately affects forest growth rates, varying with wood density, soil fertility, and water availability in different Amazonian forests (*Rozendaal et al., 2020*). The negative effect of competition on growth is stronger on forests with high wood density and water availability, but decreases with higher soil fertility. Locally, competition has a stronger effect on plateaus than valleys in Central Amazonia, as trees with the same height: diameter ratio have larger crowns in valleys (*Alencar, de Castilho & Costa, 2023*). Additionally, valleys in Central Amazonia experiencing higher tree mortality due to poor soil structure (*Toledo et al., 2011*, *2012*), and frequent gap openings may reduce competition effects, allowing the establishment of shorter trees. Therefore, forest types with contrasting soil conditions, like terra-firme and várzea, are expected to have different tree allometric relationships due to competition.

Height measurement is subject to errors from multiple sources, including method, terrain, vegetation density, crown position, and tree height, as it is difficult to identify crown top of tall trees and obtaining accurate height measurements (*Hunter et al., 2013*; *Larjavaara & Muller-Landau, 2013*; *Pereira et al., 2019*). Height measurement errors are largest in taller-than-average trees, causing biomass estimation errors of 16% at the individual level and 6% at the plot-level (*Hunter et al., 2013*). Large trees, with diameter at breast height (DBH) greater than 80 cm, show substantial height variability (32 to 71 m in the database examined by *Feldpausch et al. (2012)*), which increases the uncertainties in height-diameter allometry. The effect of height inclusion on biomass estimates varies by diameter class, with large diameter trees showing a more pronounced downward in biomass (*Feldpausch et al., 2012*). The incorporation of tree height resulted in a downward by ~4% to 11% in biomass estimates across Amazonia (*Nogueira et al., 2008b*). However, this effect varies according to forest type, with smaller trees in southern Amazon open forests leading to greater biomass overestimation per area (*Nogueira et al., 2008a*, *2008b*). Further, incorporating height reduces biomass estimates for tropical forests worldwide by 13%, though this reduction varies widely between tropical forests (*Feldpausch et al., 2012*). The inclusion of climatic variables improved large-scale height allometry and biomass estimates (*Chave et al., 2014*), although climatic models tend to overestimate height for

shorter forests (*Barbosa et al., 2019*). Therefore, further research on patterns of height-diameter relationships in different forest types is essential to improve sampling efficiency and biomass estimates, especially where local biomass equations are not available (*Sullivan et al., 2018*).

Although several models and methods exist for estimating tree height (*Castaño-Santamaría et al., 2013*; *Feldpausch et al., 2011*; *Pereira et al., 2019*), non-linear models may best capture height-diameter relationships (*Batista, Do Couto & Marquesini, 2001*; *Hess et al., 2014*; *Fayolle et al., 2016*; *De Souza, Dos Santos & Souza, 2017*; *Sullivan et al., 2018*; *Barbosa et al., 2019*; *Nascimento et al., 2020*). Height-diameter relationships vary across forest types, mainly due to differences in species composition, tree density, crown size, age, and presence of lianas (*Scaranello et al., 2012*; *Dias et al., 2017*; *Fayolle et al., 2016*; *Siliprandi et al., 2016*). Therefore, a systematic sample of tree height and diameter must be conducted to support best-fitting model selection (*Sullivan et al., 2018*).

Tree height measurements are increasingly included in forest inventory protocols (*Feldpausch et al., 2011*; *Larjavaara & Muller-Landau, 2013*; *Fayolle et al., 2016*; *Sullivan et al., 2018*), providing an opportunity to improve the modeling of height-diameter relationship across forest types. *Feldpausch et al. (2012)* proposed height-diameter equations for different Amazon regions (*e.g.*, Guyana shield), while *Chave et al. (2014)* developed a pantropical equation to estimate height using diameter and an index of environmental stress for improved height-diameter allometry in tropical forests. However, height variability across Amazonian forest types is a source of uncertainty regarding the applicability of both regional and pantropical allometric models, which require testing (*Sullivan et al., 2018*; *Barbosa et al., 2019*).

Northeastern Amazonian forests are notably tall (*Gorgens et al., 2021*). Non-flooded terra-firme forests cover most of the Guyana shield and white-water floodplain várzea forests span hundreds of kilometers along the Amazon River. These forests differ markedly in structure and species composition. Recent inventories measuring diameter and height provide an opportunity to (1) determine how height-diameter relationships vary between two Amazon forest types, (2) identify the best local height models and evaluate their performance relative to regional (*Feldpausch et al., 2012*) and pantropical (*Chave et al., 2014*) models, (3) assess the effect of species composition on height-diameter allometry, and (4) determine how the inclusion of height estimates from local, regional and pantropical models affects the estimates of biomass at different spatial scales.

# MATERIALS AND METHODS

Portions of this text were previously published as part of a thesis (*Baia, 2018*: https://ppbio.inpa.gov.br/sites/default/files/Baia_A_L_P_Dissertacao_2018.pdf).

## Study sites

This study was conducted in two different types of forest physiognomies in Northeastern Amazonia—a non-flooded terra-firme forest located at the National Forest of Amapá (NFA) and a seasonally-flooded várzea forest located in the left bank of the Amazon River.

The study site in the terra-firme forest is located in the south part of NFA (0°59′24″N and 51°38′13″W) spanning an area of 4,598,672 km$^2$ which is managed by the Chico Mendes Institute for Biodiversity Conservation (ICMBio). The climate is equatorial hot-humid, Af in Köppen's classification (*Kottek et al., 2006*). Temperature varies between 22 °C and 32 °C and annual rainfall is around 2,284 mm with a rainy season peak between February and May and a dry season peak between August and November (*Instituto Chico Mendes de Conservação da Biodiversidade (ICMBio), 2014*). Topography is slightly undulated with 100–200 m a.s.l. elevation and an inclination of around 7% on slopes. Ultisols are predominant and fertility is low (*Quesada et al., 2011*; *Instituto Chico Mendes de Conservação da Biodiversidade (ICMBio), 2014*). Old-growth terra-firme forest presents a canopy high of around 30 m with frequent emergent trees reaching 50 m height.

The estuarine várzea forest spans an area of approximately 13,189 km$^2$ (00°06′54″S and 51°17′20″W) in the municipality of Mazagão, south of Amapá State, Brazil. The climate is Am-type, super-humid equatorial (*Kottek et al., 2006*), with an annual average temperature of 27.6 °C and annual rainfall averaging 2,531 mm (*INMET, 2017*). Rainfall is concentrated between February and May and decreases sharply in the dry season from August to November. Topography is flat with elevation varying from 1 to 30 m a.s.l.. The flooding regime follows the tidal cycle, which reaches ~2 m height in the end of the rainy season (*Furtado et al., 2023*). Haplic gleysoils are predominant with high silt content and fertility is relatively high (*Quesada et al., 2011*). The canopy height varies from 20 to 25 m, with emergent trees reaching 40 m height.

## Step-by-step methodological approach

We applied a four-step methodological approach to assess how the incorporation of tree height estimated with local height-diameter equations, compared to two published models, can improve the estimates of biomass at different spatial scales in the Northeastern Amazon: at a local scale encompassing a non-flooded terra-firme and a seasonally-flooded várzea forest site, and at regional scale, combining data of both sites. First, we measured the diameter and height of 1,962 trees in both forests using a highly accurate hypsometer. Second, we used metrics of precision, accuracy and parsimony to select the best models from a set of 10 gathered from the forest science literature, to provide local height-diameter equations. Third, we used these best local models and two previously published height models (the Guyana shield model: *Feldpausch et al. (2012)* and the pantropical model: *Chave et al. (2014)*) to estimate tree height in our dataset, and compared these estimates to measured heights assessing errors associated to accuracy and precision. Finally, we calculated tree biomass using equations that included and excluded height, developed for each forest type (using models by *Lima (2015)* for terra-firme and *Chave et al. (2005)* for várzea) and a regional scale combining the data of both forests (using *Chave et al., 2014*). We compared these biomass estimates to those estimates using directly measured height, by assessing the errors associated to accuracy and precision. A detailed description of the procedures and equations is provided below.

## Sampling design and tree measurements

In the terra-firme site, five 0.5-ha plots (40 × 125 m), at a distance of at least 1 km from each other, were used in this study. These plots were previously established in 2008 along two east-west trails and a complete inventory was carried out from 2015 to 2016. Trees were inventoried using a size-based stratification scheme: trees with diameter at breast height (DBH) ≥1 cm were inventoried in subplots of 0.0125 ha (1 × 125 m); trees with DBH ≥10 cm were sampled in subplots of 0.25 ha (20 m × 125 m); and trees with DBH ≥ 30 cm were inventoried in the entire plot (40 × 125 m). Four 0.5-ha permanent plots (50 × 100 m), located at least 1 km from each other, were established in the várzea forest along the upstream of the Mazagão River (a small tributary of the Amazon River) in 2011 (*Dantas, 2015*), in which all trees with ≥5 cm DBH were inventoried. A distance of 1 km between plots was used to maintain a minimum level of independence between sample units.

In both terra-firme and várzea sites, trees were mapped and tagged with aluminum numbered tags. The DBH was measured at 1.3 m above the ground and 0.5 m above irregularities or buttresses using a fabric diameter tape (Forestry Suppliers, model 283D) for trees ≥5 cm DBH and with a digital caliper for smaller trees. Total tree height was defined as the distance from the base of the trunk up to the top of the crown. Height measurements were conducted using a TruPulse® 360 laser hypsometer (Laser Technology Inc., Centennial, CO, USA) and were calculated using the tangent method (see *Larjavaara & Muller-Landau, 2013*). Only one of us (A.L.P. Baia) took the height measurements with the hypsometer in the field to avoid introducing operator-associated variability.

Trees were identified by an experienced parataxonomist and botanical vouchers were collected for comparison with herbarium material when identification was not possible. Fertile vouchers were deposited in the herbarium of the National Institute for Amazon Research. The permission for botanical sampling was granted by the Instituto Chico Mendes de Conservação da Biodiversidade—ICMBio (permit number: 510561).

## Model selection for height-diameter allometry

To find a best-fit allometric model to describe the relationship between height and diameter, we used a set of 10 models (Table 1) and tested their fit to the data of terra-firme and várzea forests separately as well as the pooled dataset.

Model parameters were estimated using iterative non-linear regression (*Bailey, 1980*). The statistics for model selection were calculated as following (*Burnham & Anderson, 2002*; *Chave et al., 2005*):

$$\text{AIC} = 2k - 2ln(L) \tag{1}$$

where AIC is the Akaike Information Criterion, $K$ is the number of parameters estimated in the model, and $L$ is the value of maximum likelihood estimated for the model;

$$R^2_{adj} = R^2 - \left(\frac{K-1}{N-K}\right) \cdot \left(1 - R^2\right) \tag{2}$$

where $R^2_{adj}$ is the multiple adjusted coefficient of determination and $N$ is the sample size.

**Table 1 Allometric models used to describe the relationship between total height and diameter of trees from terra-firme and várzea forests in northeastern Amazonia.**

| Model name | Equation | Parameter interpretation |
|---|---|---|
| Quadratic | $H = exp(a + b \times ln(DBH) + c \times ln(DBH)^2) + \varepsilon$ | $a$ = average height when diameter is zero |
|  |  | $b$ = change in height for each unit change in diameter |
|  |  | $c$ = represents the quadratic relation between height and diameter |
| Michaelis-Menten | $H = a \times DBH/(b + DBH) + \varepsilon$ | $a$ = maximum height |
|  |  | $b$ = diameter that is needed to achieve a height that is half the maximum height |
| Weibull | $H = a \times (1 - exp(-b \times DBH^c)) + \varepsilon$ | $a$ = maximum height |
|  |  | $b$ = rate of change |
|  |  | $c$ = shape |
| Three parameter exponential | $H = a - b \times exp(-c \times DBH) + \varepsilon$ | $a$ = maximum height |
|  |  | $b$ = height range |
|  |  | $c$ = rate of change |
| Two parameter exponential | $H = a \times (1 - exp(-b \times DBH)) + \varepsilon$ | $a$ = maximum height |
|  |  | $b$ = rate of change |
| Gompertz | $H = a \times exp (-b \times exp(-c \times DBH)) + \varepsilon$ | $a$ = maximum height |
|  |  | $b$ = a rate that multiplied by $a$ parameter returns the height when diameter is zero |
|  |  | $c$ = rate of change |
| Power function | $H = a \times DBH^b + \varepsilon$ | $a$ = constant value of height |
|  |  | $b$ = rate of change |
| Modified two parameter exponential | $H = a \times exp(b/DBH) + \varepsilon$ | $a$ = maximum height |
|  |  | $b$ = rate of change |
| Logistic | $H = a/(1 + b \times exp(-c \times DBH)) + \varepsilon$ | $a$ = maximum height |
|  |  | $b$ = height range |
|  |  | $c$ = rate of change |
| Log-linear | $H = a + b \times ln(DBH) + \varepsilon$ | $a$ = average height when diameter is zero |
|  |  | $b$ = change in height for each unit change in diameter |

**Note:**
DBH is the diameter at breast height (cm) measured at 1.3 m above the ground, H is the total tree height (m), $a$, $b$ and $c$ are equation parameters, $ln$ is the natural logarithm, $exp$ is the antilogarithm and $\varepsilon$ is the error.

The *AIC* penalizes the models according to the number of parameters estimated and was employed to rank the models from the best (smallest *AIC*) to the worst (highest *AIC*) fits (*Burnham & Anderson, 2002*) and $R^2_{adj}$ provides the quantity of variation explained by a model after a penalization by the number of parameters estimated in the model. We also used the residual standard error (*RSE*) to assess accuracy, and ranked the models firstly by higher values of the adjusted $R^2$, lower *RSE* and lower *AIC* values. Additionally we evaluated the fit of the best models regressing the observed against the predicted values. Subsequently, we applied Graybill's test (*Leite & Oliveira, 2002*) to test the hypotheses that the intercept differs from zero and the slope differs from one. We also used Theil's error decomposition (*Smith & Rose, 1995*) to separate the sum of squared residuals into: model lack-of-fit to test for the overall model fit; no bias to test if the intercept differs from zero;
consistency, to test if the slope differs from one; and the regression lack-of-fit to identify non-linear deviations.

## Taxonomic effect on height-diameter allometry

We included species identity as a random factor in the linear height ($H$)-diameter ($D$) models (Log-linear: $H = a + b \times ln(D) + 1|species$ and Quadratic: $H = a + b \times ln(D) + c \times ln(D)^2 + 1|species\ identity$) using generalized linear mixed models (GLMM) to estimate the contribution of taxonomic variation to the models' precision. The marginal ($R^2_m$) and conditional ($R^2_c$) $R^2$ were calculated for each model (*Nakagawa & Schielzeth, 2013*) to separate the contribution of fixed from random factor since the $R^2_m$ represents the contribution of the fixed factor and the $R^2_c$ represents the variation explained by both fixed and random factors.

We tested for the phylogenetic autocorrelation by calculating the phylogenetic signal associated with the traits (diameter and height), and applied the phylogenetic generalized least squares (PGLS) using estimates of maximum height and maximum diameter per taxon. We used the best height-diameter model chosen in the selection procedure, and applied PGLS to estimate the parameters of the relationship between maximum height ($H_{max}$) and maximum diameter ($D_{max}$) ($H_{max} = a + b \times ln(D_{max}) + c \times ln(D_{max})^2 + \Sigma$, where $\Sigma$ is the phylogenetic covariance matrix calculated on branch lengths of the phylogenetic tree; Appendix S1). Further, we compared the performance of this model (using RSE, adjusted pseudo $R^2$ and AIC) to the model run with least squares regression without accounting for phylogenetic structure.

## Height estimation at local and regional scales

We used the best-fit models for each forest type and the pooled dataset to estimate tree height. Tree height was also estimated using two available models in the literature—the regional model for Guyana shield forests (*Feldpausch et al., 2012*) and the pantropical climate model (*Chave et al., 2014*), respectively:

$$H = 42.845 \times (1 - exp(-0.0433 \times D^{0.9372})) \tag{3}$$

$$H = exp(0.893 - E + 0.760 \times ln(D) - 0.0340 \times ln(D)^2) \tag{4}$$

where $H$ is the total height in meters, $D$ is the DBH in centimeter measured at 1.3 m above the ground, $ln$ is the natural logarithm, $exp$ is the antilogarithm and $E$ is an index of environmental stress which incorporates measures of temperature seasonality, climatic water deficit and precipitation seasonality. This index describes variation in climatic constraints of tree growths and is able to improve the adjustment of height-diameter model (*Chave et al., 2014*).

## Biomass estimation using estimated tree heights

Aboveground biomass ($AGB$) was estimated using equations with and without height developed for different forest types as described below. Since the log-transformation is expected to underestimate the $AGB$, we used a correction factor [$CF = exp(RSE^2/2)$] based

on the residual standard error (*RSE*) of the regression which was multiplied by the estimated *AGB* (*Baskerville, 1972*). For the terra-firme forest we used equations with trees sampled in a terra-firme site within the Amapá State Forest, located 65 km southwest apart (*Lima, 2015*):

$$AGB = exp(-3.04385 + 0.94863 \times ln(D^2 \times H)) \times 1.08071 \tag{5}$$

$$AGB = exp(-1.91172 + 2.45043 \times ln(D)) \times 1.094052 \tag{6}$$

For the várzea forest we used equations proposed for tropical moist forests and wet mangroves (*Chave et al., 2005*), respectively:

$$AGB = -3.027 + ln(\rho \times D^2 \times H) \times 1.051195 \tag{7}$$

$$AGB = \rho \times exp(-1.349 + 1.980 \times ln(D) + 0.207 \times (ln(D))^2 - 0.0281 \times (ln(D))^3) \times 1.065419 \tag{8}$$

For both forests we used equations proposed by *Chave et al. (2014)* as follow:

$$AGB = 0.0673 \times (\rho \times D^2 \times H)^{0.976} \tag{9}$$

$$AGB = exp(-1.8030 - 0.976 \times E + 0.976 \times ln(\rho) + 2.673 \times ln(D) - 0.0299 \times ln(D)^2) \times 1.089027 \tag{10}$$

where $\rho$ is the wood density (g cm$^{-3}$) which was obtained from the global database available at https://datadryad.org/dataset/doi:10.5061/dryad.234 (*Zanne et al., 2009*). We used the average wood density for species level when available, and for trees identified at either genus or family level, $\rho$ was calculated as the mean of genus or family, respectively, and for unidentified individuals, an average $\rho$ calculated for the plot was used. Biomass was calculated for individuals ≥5 cm DBH.

## Precision and accuracy of estimated height and biomass

We compared the measured height to estimated heights from local equations developed in this study as well as published equations from *Feldpausch et al. (2012)* and *Chave et al. (2014)*. Additionally, we compared biomass calculated using measured height to biomass calculated using estimated height (using height models from this study and previously published) and to biomass calculated without height (using diameter only). For these comparisons, we quantified the total error (TE), systematic error (SE), and random error (RE). Total error was evaluated based on the root mean square error:

$$TE = \sqrt{\frac{1}{n}\sum(h_{est,i} - h_{meas,i})^2} \tag{11}$$

where $h_{meas,i}$ is the measured (actual) height of the $i^{th}$ tree, $h_{est,i}$ is the estimated height of the $i^{th}$ tree and $n$ is the number of trees. We quantified systematic error as the mean measurement error:

$$SE = \frac{1}{n}\sum(h_{est,i} - h_{meas,i}) \tag{12}$$

We quantified random error as the sample standard deviation of the measurement errors:

$$R = \sqrt{\frac{1}{n-1}\sum (h_{est,i} - h_{meas,i} - SE)^2} \qquad (13)$$

Since errors increased with the true height (or biomass), we also calculated all of the above in proportional terms. Specifically, proportional total error was calculated as follows:

$$CVTE = \sqrt{\frac{1}{n}\sum \left(\frac{h_{est,i} - h_{meas,i}}{h_{meas,i}}\right)^2} \qquad (14)$$

proportional systematic error as follows:

$$CVSE = \frac{1}{n}\sum \left(\frac{h_{est,i} - h_{meas,i}}{h_{meas,i}}\right) \qquad (15)$$

and proportional random error as follows:

$$CVRE = \sqrt{\frac{1}{n-1}\sum \left(\frac{h_{est,i} - h_{meas,i}}{h_{meas,i}} - CVSE\right)^2} \qquad (16)$$

Note that in general, higher precision is defined by lower random error, and higher accuracy is defined as lower systematic error.

## RESULTS

We sampled 1,156 trees of 367 species and 148 genera from 48 botanical families in the terra-firme sites, and 806 trees of 65 species and 59 genera from 29 botanical families in várzea (Table S1). A total of 90.2% of individuals were identified up to species level, 98.8% up to genus level, and 0.8% were not identified. The most abundant species in terra-firme were *Vouacapoua americana* (3.64%), *Eschweilera coriaceae* (3.64%) and *Lecythis chartacea* (2.95%), and the most abundant in várzea were *Mora paraensis* (35.98%), *Pentaclethra macroloba* (8.93%) and *Patinoa paraensis* (4.84%).

In the terra-firme sites, trees ranged from 1 to 109.1 cm DBH and 1.2 to 51.9 m in total height. In várzea sites, trees ranged from 5 to 139.5 cm DBH and 2.3 to 38.8 m in total height. The mean DBH in terra-firme excluding trees smaller than 5 cm DBH (24.5 ± 16.8 cm; mean ± standard deviation) was higher than in várzea (22.4 ± 18.6 cm) ($t = 2.42$, $p = 0.02$), likewise mean total height (22.9 ± 8.8 m in terra-firme *vs.* 15.8 ± 7 m in várzea; $t = 17.8$, $p < 0.001$). However, mean wood density was slightly lower in terra-firme (0.70 ± 0.14 g.cm$^{-3}$) than in várzea (0.73 ± 0.14 g.cm$^{-3}$) ($t = -5.3$, $p < 0.001$).

### Height-diameter allometry

All the models provided suitable fits for terra-firme forest, but to a lesser extent when data from both of terra-firme and várzea were combined, and least of all for the várzea forest. For example, the Quadratic model was the best fitting model for both terra-firme and várzea forests, but $R^2$, *RSE* and *AIC* were highly different between the two forest physiognomies. On the other hand, Weibull and Michaelis-Menten were the best-fitted

model in the rank for both forests together, but with an intermediate fit between terra-firme and várzea (Table 2). However, for the three cases, model performances differed greatly for large trees (Fig. 1).

The relationship of the observed with predicted values produced intercepts and slopes that significantly ($F > 80$, $p < 0.001$) deviated from zero and one, respectively (Table S2). The model lack-of-fit was not significant ($p \geq 0.05$) for models in terra-firme, but there were significants ($p < 0.001$) lack-of-fit for várzea and the combined forest data. None of the models showed significant deviations of the intercepts from zero ($p$ for no bias >0.69) or of the slopes from one ($p$ for consistency ≥0.52), indicating a symmetrical relationship (1:1 passing through zero) between observed and predicted values. Indeed, none of the three best models (Quadratic, Michaelis-Menten and Weibull) deviated from an 1:1 relation between the observed with the predicted height (Fig. 2). The regression lack-of-fit was not significant ($p > 0.05$) for two models (Quadratic and Michaelis-Menten) in terra-firme, but there were significant deviations for Weibull model ($p = 0.047$) in terra-firme and for all models ($p < 0.001$) in várzea and for pooled data of both forests, indicating non-linear deviations.

The inclusion of species identity as a random factor improved the models' fit by only 2% for terra-firme, 5% for várzea and 10% for both forests (Table S3), indicating that taxonomic variation has little impact on height-diameter allometry within forest types, but the contribution increased for models fitted with pooled data of both forests.

The results indicated a weak phylogenetic signal for traits and small reduction of models fit after controlling for phylogenetic structure (Table S4). The best height-diameter model (Quadratic) lost only 1% of precision after controlling for phylogenetic autocorrelation, indicating the phylogenetic structure did not play an important role in trait relationships for both terra-firme or várzea forests.

## Precision and accuracy of height-diameter models

Random error (a measure of precision) represented most of total error associated to estimates of height (97 ± 8%, 95 ± 13% and 99 ± 4% for terra-firme, várzea and both forests, respectively) and biomass (98 ± 4%, 98 ± 6%, and 99 ± 3%) (see Tables S5 and S6 for estimates of all error types). Therefore, to avoid redundancy we described the results for total and systematic errors only. Overall, irrespective of size class, both height and biomass estimated with heights predicted from the best local models presented lower total and systematic errors in comparison to height and biomass estimated from regional (Guyana shield) and pantropical height models and biomass estimated without height (Fig. 3; Tables S5 and S6).

For terra-firme forest, the Guyana shield model performed similarly to local models, but the pantropical height model inflated the total error by more than one third (Fig. 3A). In contrast, for várzea, the Guyana shield model increased the total error by more than one third, whereas the pantropical model performed similarly to local models (Fig. 3B). For both forests combined, neither the Guyana shield nor the pantropical models differed from local models within size class (Fig. 3C). The systematic errors mirrored the total errors (Figs. 3D–3F), witht the Guyana shield model overestimating height for várzea

**Table 2 Results of model selection for the height-diameter allometry for two forest types (terra-firme and várzea) in northeastern Amazonia.**

| Model name | Parameter | Estimate | SE | RSE | Adj. pseudo $R^2$ | AIC |
|---|---|---|---|---|---|---|
| **Terra-firme** | | | | | | |
| Quadratic | a | 0.63296 | 0.081 | 4.018 | 0.87 | 6,501.2 |
| | b | 1.11213 | 0.052 | | | |
| | c | −0.09504 | 0.008 | | | |
| Michaelis-Menten | a | 51.09644 | 0.988 | 4.021 | 0.87 | 6,501.9 |
| | b | 25.11112 | 0.99 | | | |
| Weibull | a | 42.84443 | 1.478 | 4.031 | 0.87 | 6,508.6 |
| | b | 0.05836 | 0.003 | | | |
| | c | 0.84815 | 0.025 | | | |
| Three par. Exponential | a | 38.97337 | 0.659 | 4.062 | 0.87 | 6,526.4 |
| | b | 37.80493 | 0.604 | | | |
| | c | 0.04184 | 0.002 | | | |
| Two par. Exponential | a | 37.81206 | 0.539 | 4.089 | 0.87 | 6,540.8 |
| | b | 0.0465 | 0.001 | | | |
| Gompertz | a | 34.67709 | 0.411 | 4.294 | 0.86 | 6,654.5 |
| | b | 2.28522 | 0.053 | | | |
| | c | 0.08633 | 0.003 | | | |
| Power function | a | 4.14876 | 0.119 | 4.315 | 0.86 | 6,665.1 |
| | b | 0.54281 | 0.008 | | | |
| Modified two par. Exponential | a | 39.41123 | 0.49 | 4.572 | 0.86 | 6,798.7 |
| | b | −9.87593 | 0.249 | | | |
| Logistic | a | 32.81866 | 0.345 | 4.54 | 0.84 | 6,783.5 |
| | b | 6.2464 | 0.3 | | | |
| | c | 0.13849 | 0.004 | | | |
| Log-linear | a | −2.70755 | 0.294 | 4.585 | 0.84 | 6,805.0 |
| | b | 8.65375 | 0.113 | | | |
| **Várzea** | | | | | | |
| Quadratic | a | 0.51504 | 0.158 | 4.161 | 0.65 | 4,590.7 |
| | b | 1.07489 | 0.1 | | | |
| | c | −0.10005 | 0.015 | | | |
| Michaelis-Menten | a | 33.09269 | 0.811 | 4.164 | 0.65 | 4,590.9 |
| | b | 18.42734 | 1.032 | | | |
| Weibull | a | 29.80081 | 1.633 | 4.163 | 0.65 | 4,591.2 |
| | b | 0.085 | 0.006 | | | |
| | c | 0.76603 | 0.047 | | | |
| Three par. Exponential | a | 27.88745 | 0.823 | 4.172 | 0.64 | 4,594.9 |
| | b | 24.37061 | 0.674 | | | |
| | c | 0.03977 | 0.004 | | | |
| Two par. Exponential | a | 25.68178 | 0.46 | 4.225 | 0.64 | 4,614.4 |
| | b | 0.05765 | 0.002 | | | |
| Gompertz | a | 26.80285 | 0.634 | 4.2 | 0.64 | 4,605.6 |

| Model name | Parameter | Estimate | SE | RSE | Adj. pseudo $R^2$ | AIC |
|---|---|---|---|---|---|---|
| | b | 1.53882 | 0.058 | | | |
| | c | 0.05893 | 0.004 | | | |
| Power function | a | 4.45246 | 0.182 | 4.279 | 0.63 | 4,634.6 |
| | b | 0.43177 | 0.012 | | | |
| Modified two par. Exponential | a | 27.50285 | 0.424 | 4.285 | 0.63 | 4,636.8 |
| | b | −8.3306 | 0.268 | | | |
| Logistic | a | 26.19871 | 0.547 | 4.231 | 0.63 | 4,617.6 |
| | b | 2.85977 | 0.16 | | | |
| | c | 0.07903 | 0.005 | | | |
| Log-linear | a | −5.41997 | 0.574 | 4.165 | 0.65 | 4,591.0 |
| | b | 7.52835 | 0.197 | | | |
| **Both forests** | | | | | | |
| Quadratic | a | 0.48064 | 0.092 | 4.937 | 0.75 | 11,838.5 |
| | b | 1.16793 | 0.059 | | | |
| | c | −0.10905 | 0.009 | | | |
| Michaelis-Menten | a | 42.1399 | 0.764 | 4.935 | 0.75 | 11,835.8 |
| | b | 21.77561 | 0.849 | | | |
| Weibull | a | 35.07114 | 1.032 | 4.934 | 0.75 | 11,836.6 |
| | b | 0.0661 | 0.003 | | | |
| | c | 0.85189 | 0.028 | | | |
| Three par. Exponential | a | 32.91484 | 0.529 | 4.946 | 0.75 | 11,845.9 |
| | b | 31.48734 | 0.487 | | | |
| | c | 0.04521 | 0.002 | | | |
| Two par. Exponential | a | 31.8906 | 0.421 | 4.969 | 0.74 | 11,862.6 |
| | b | 0.0514 | 0.001 | | | |
| Gompertz | a | 30.33256 | 0.367 | 5.059 | 0.73 | 11,934.4 |
| | b | 2.06328 | 0.052 | | | |
| | c | 0.08256 | 0.003 | | | |
| Power function | a | 4.28977 | 0.12 | 5.167 | 0.72 | 12,016.2 |
| | b | 0.49388 | 0.008 | | | |
| Modified two par. Exponential | a | 33.82038 | 0.378 | 5.204 | 0.73 | 12,043.9 |
| | b | −9.20394 | 0.21 | | | |
| Logistic | a | 29.21444 | 0.316 | 5.189 | 0.72 | 12,033.6 |
| | b | 4.83673 | 0.215 | | | |
| | c | 0.12164 | 0.004 | | | |
| Log-linear | a | −2.95482 | 0.304 | 5.228 | 0.72 | 12,062.3 |
| | b | 7.79326 | 0.111 | | | |

**Note:**
Models are ranked from the best to the worst according to adjusted pseudo $R^2$, residual standard error (RSE) and Akaike Information criterion (AIC). The a, b and c are the parameters for model equations described in Table 1 and SE is the standard error of the estimate.

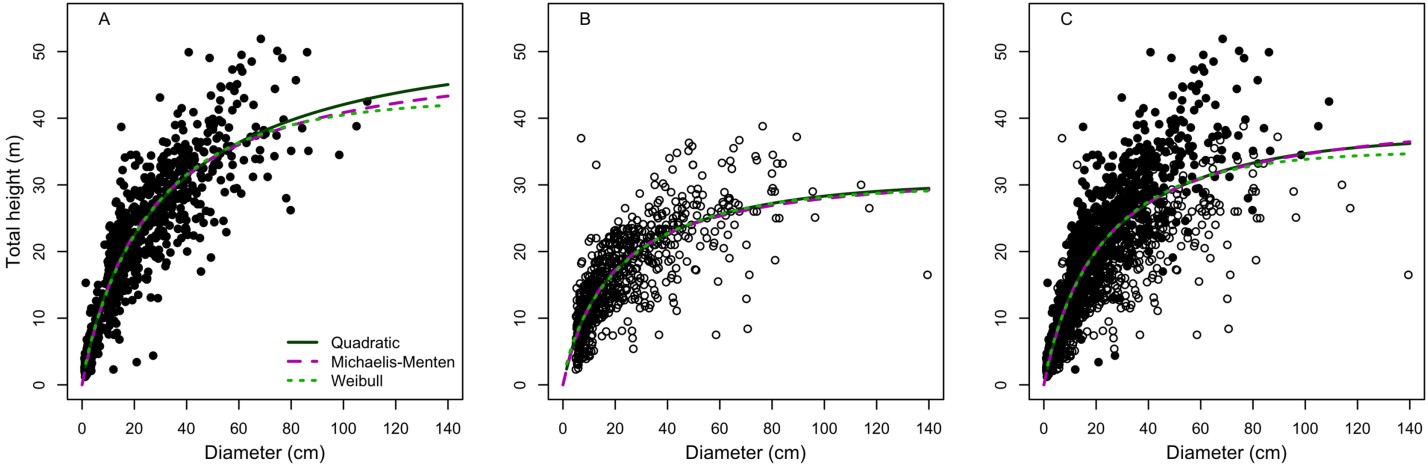

**Figure 1 Model fitting to best three height-diameter relationships for (A) terra-firme, (B) várzea and (C) pooled data of both forests in northeastern Amazonia.** Parameter estimates for the Quadratic, Michaelis-Menten and Weibull models are in Table 2.

(by 21%, 42% and 29% for small, large and all trees, respectively), and for both forests (by 17% and 10% for large and all trees, respectively), while the pantropical model underestimated the height by 20% across size classes for terra-firme and overestimated it in várzea by 15% for large trees and 6% for all trees.

The biomass estimates in terra-firme, calculated with heights predicted by the Guyana shield model, were similar to the estimates calculated with height from local models, but the pantropical model inflated the total error by one third (Fig. 3G). In várzea, the Guyana shield model more than doubled the error compared to local models and the pantropical model inflated the error by almost two-thirds for both large and all trees (Fig. 3H). For both forests, the Guyana shield model contributed to increase the total error by one-quarter, while the pantropical model showed similar performance to local models (Fig. 3I).

Systematic errors for biomass estimated with height from the Guyana shield model were near zero (Fig. 3J). In contrast, it led to a high overestimation (30% for small and 41% for large and all trees) in várzea (Fig. 3K) and a moderate overestimation (11% for small and 17% for large and all trees) for both forests (Fig. 3L). Conversely, the pantropical model underestimated the biomass in terra-firme (20% for small, 16% for large and 17% all trees) and overestimated it in várzea (18% for large and 17% for all trees). The model without height underestimated biomass across tree sizes in várzea, with smaller errors for terra-firme and both forests combined. The model without height underestimated the biomass for várzea (14%, 16%, and 15%, for small, large and all trees, respectively), with smaller error for terra-firme and for both forests combined. As shown by the cumulative biomass curves in Fig. 4, the pantropical height model reduced biomass estimates for terra-firme by 57 Mg ha$^{-1}$ (15%) (Fig. 4A), whereas the Guyana shield model increased it by 166 Mg ha$^{-1}$ (44%). In várzea, the pantropical model overestimated the biomass by 93 Mg ha$^{-1}$ (25%) (Fig. 4B). For both forests, the Guyana shield model overestimated the

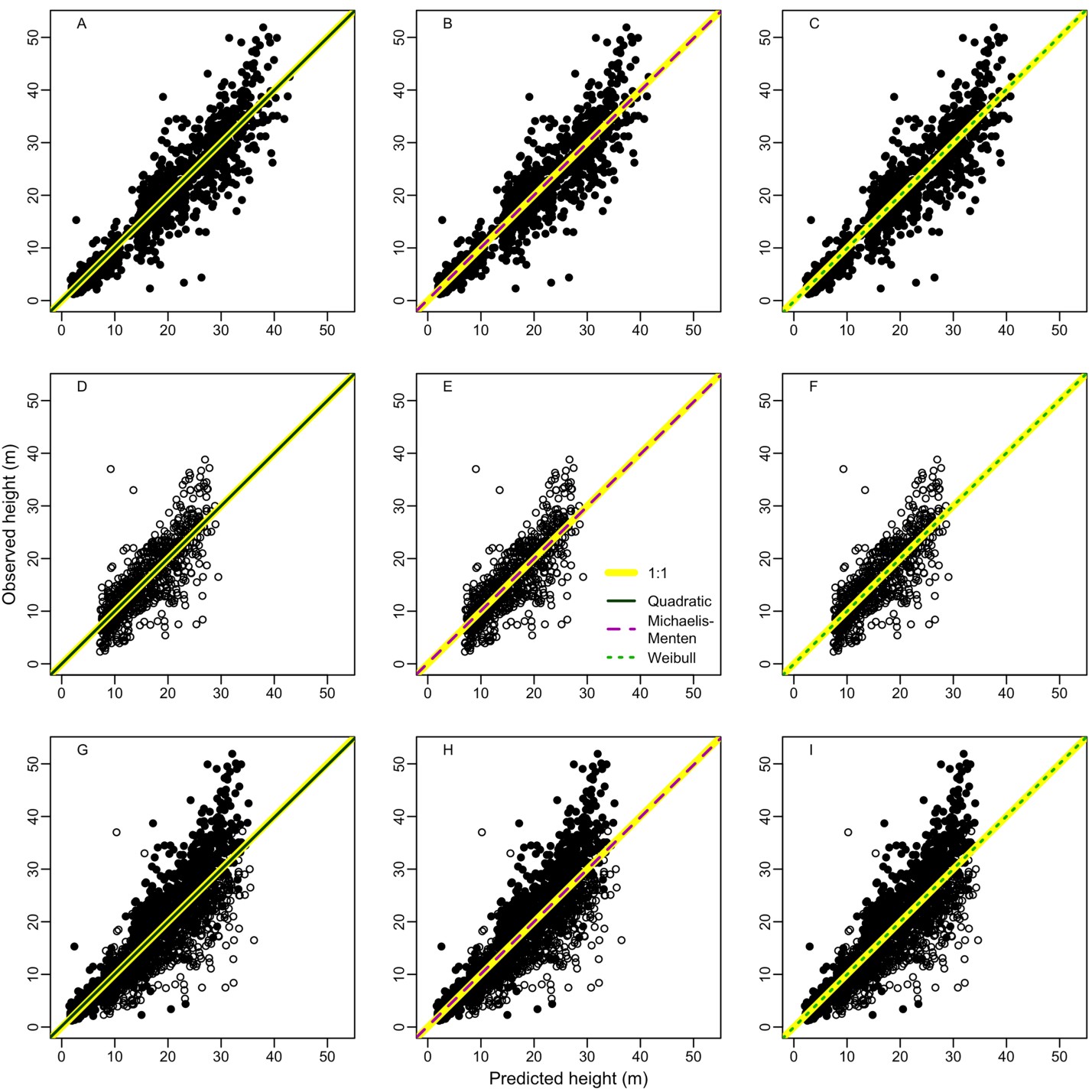

**Figure 2 Relationship between observed and predicted height of the best height-diameter models for (A–C) terra-firme, (D–F) várzea and (G–I) pooled data of both forests in northeastern Amazonia.** The straight yellow line denotes a 1:1 relation and the lines for the Quadratic, Michaelis-Menten and Weibull models denote linear regressions between observed and predicted height.

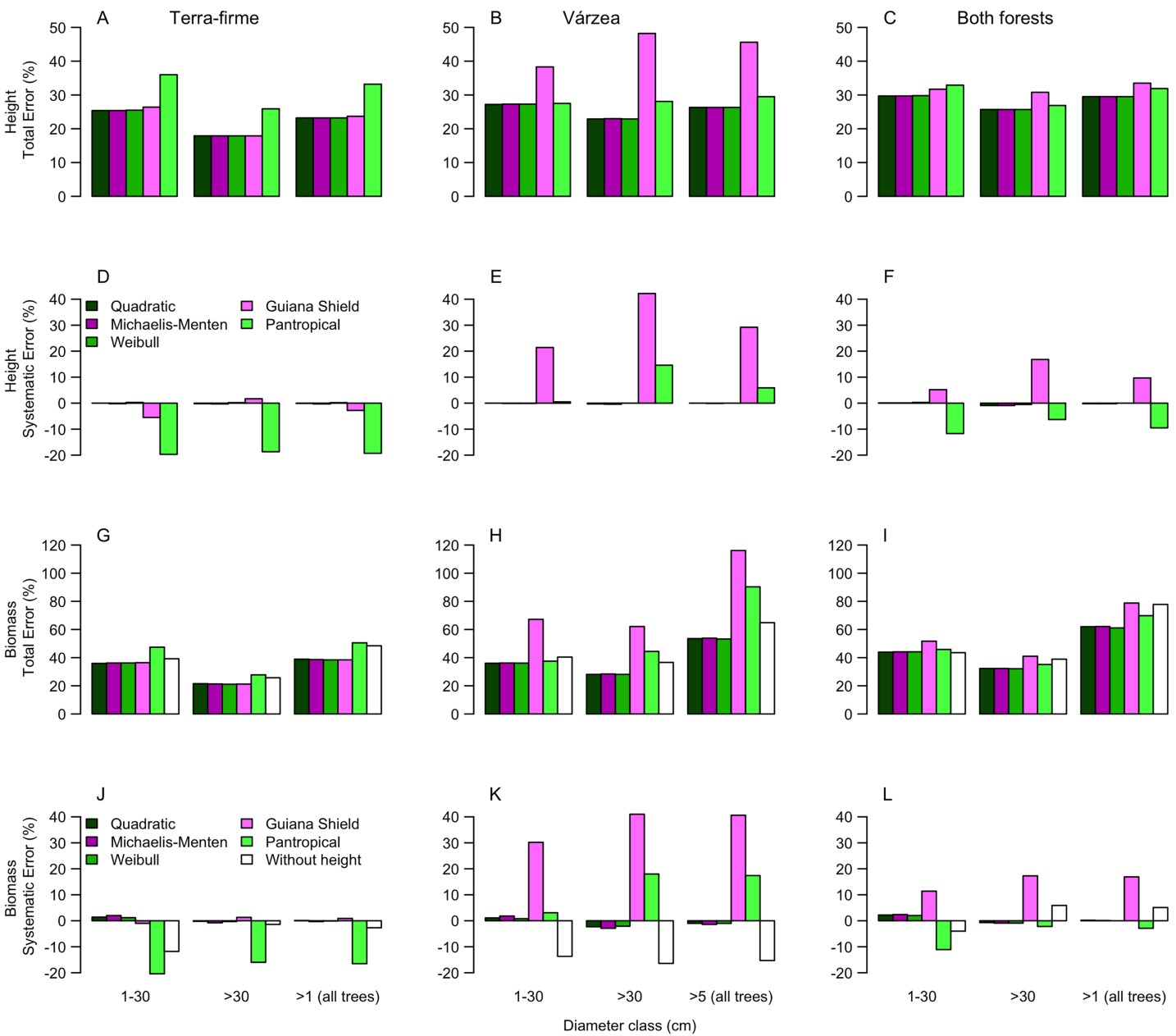

**Figure 3 Comparisons of model performance using total (%) and systematic (%) erros of the estimated height (A–F) and estimated biomass (G–L) of trees of different diameter size classes for two forest types (terra-firme and várzea) and pooled data of both forests in northeastern Amazonia.** The heights estimated with models (Quadratic, Michaelis-Menten, Weibull, Guyana Shield and Pan tropical) were compared with actual heights. Biomass estimated with and without height estimated by models were compared with biomass estimated with actual height (measured at the field). The covariances of the total and systematic errors were used as measures of precision and accuracy, respectively. Different biomass equations using height and without height were used for terra-firme (*Lima, 2015*), várzea (*Chave et al., 2005*) and both forests (*Chave et al., 2014*) (see Methods).

biomass by 102 Mg ha$^{-1}$ (21%) (Fig. 4C). Biomass estimates without height was underestimated for várzea (52 Mg ha$^{-1}$) and overestimated for both forests (69 Mg ha$^{-1}$) by 14%.

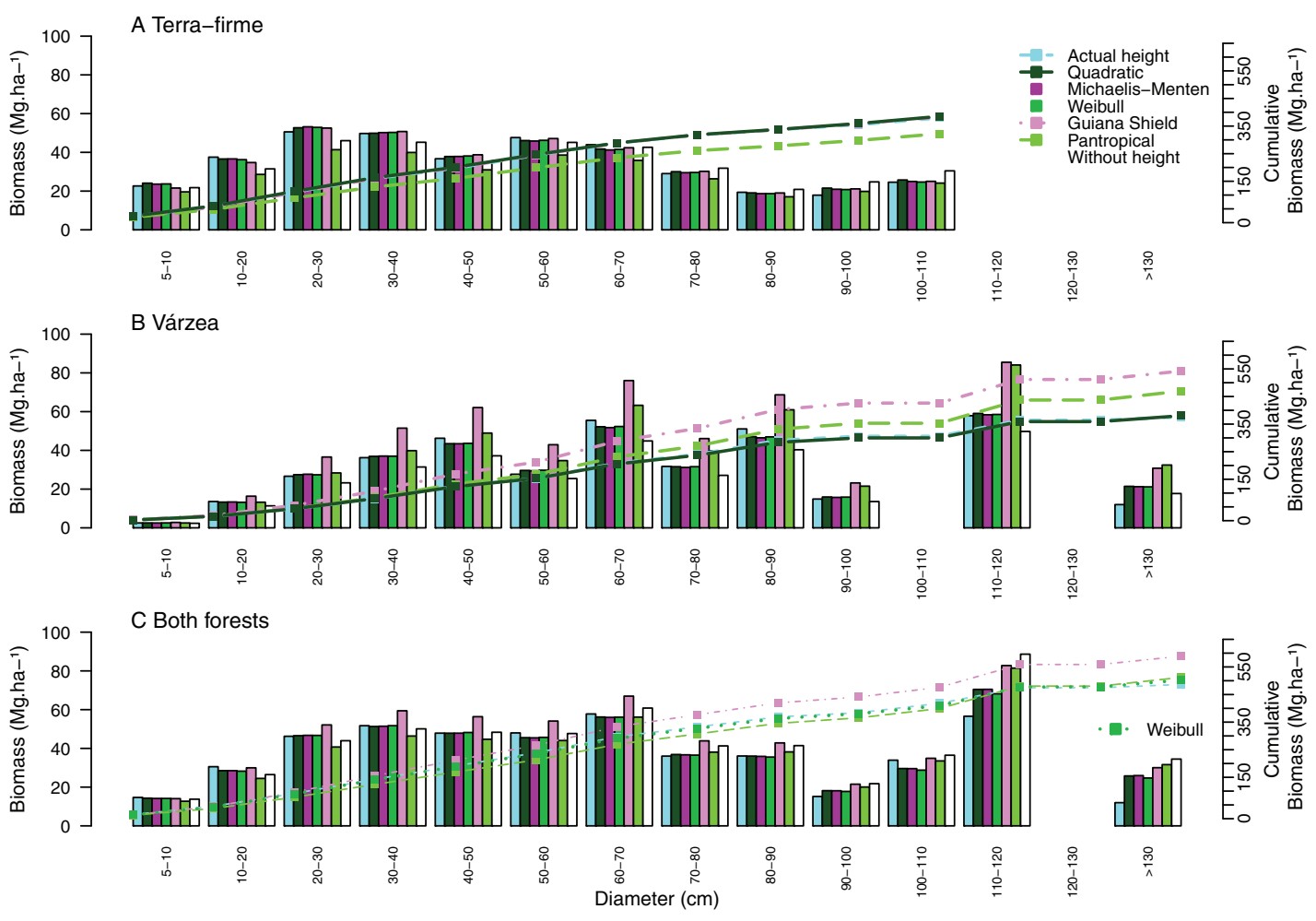

**Figure 4 Cumulative aboveground biomass across tree size classes for (A) terra-firme, (B) várzea and (C) pooled data of both forests in northeastern Amazonia.** Cumulative biomass is shown using actual heights, heights estimated from the best local model of each forest type, Guyana shield height model, Pantropical climate height model, and biomass estimated without height. Different biomass equations were used for terra-firme (*Lima, 2015*), várzea (*Chave et al., 2005*) and both forests (*Chave et al., 2014*) (see Methods).

## DISCUSSION

In this study we showed that height-diameter allometry varies between two representative forest types (non-flooded terra-firme and seasonally-flooded várzea forests) in northeastern Amazonia. We fit 10 models to estimate tree height and showed that Quadratic, Michaelis-Menten, and Weibull models fit better at local and regional scale. These models predicted a non-linear relationship where the heights of large trees stabilize or increase more slowly relative to diameter. Additionally, we demonstrated that local height models outperformed the regional (*Feldpausch et al., 2012*) and pantropical (*Chave et al., 2014*) models, and that the height estimated with local models can be used in equations to provide more accurate biomass estimates compared to biomass estimated using height from regional and pantropical models, as well as biomass estimates without height.

These local equations can be used to predict height of a wide range of tree sizes (1 to 140 cm DBH), making a relevant contribution to the allometry of large trees (≥40 cm DBH), which represent an important portion of forest biomass in Amazonia (*Nogueira et al., 2008b*) and especially in the northeastern portion of the basin because of the higher density of tall trees (>70 m) (*Gorgens et al., 2021*). We suggest that these models must be used with caution, limited to the range of tree diameter of this dataset in order to avoid underestimation of height for larger trees. Also, some variability in the height measurement, associated with the equipment (hypsometer) and tangent method, may introduce uncertainty to the model accuracy. However, a recent study (*Pereira et al., 2019*) showed that hypsometer may underestimate the total height of short trees (10–20 m) by 0.3 m and overestimate the height of tall trees (20–30 m) by 1 m, demonstrating that error associated with height measurement in the field is low. Lastly, we used a limited number of plots from each forest type, which cannot fully capture the variability in height-diameter allometry across Guiana Shield forests. Therefore, to avoid pseudo-replication, our conclusions are restricted to a specific part of terra-firme and várzea forests in the region.

## Variation in tree height allometry between forest types

The best models we selected in this study showed a better fit for trees from terra-firme compared to várzea. These variations in allometry may be associated differences in tree stand and edaphic characteristics between forests. Trees of terra-firme are expected to be taller than trees in várzea at the same diameter. In fact, trees <30 cm DBH and trees ≥30 cm DBH were 6 and 8 m taller in terra-firme (median = 19 and 32 m) than in *várzea* forest (13 and 24 m), respectively. The terra-firme forests have deep, clayey, well-drained, and nutrient-poor soils (*Quesada et al., 2011*), providing good support to trees grow slow and tall (*Gorgens et al., 2021*). In contrast, várzea forests grow on alluvial soils formed by Andean sediments, which are generally shallow and rich in nutrients (*Quesada et al., 2011*), and frequent waterlogging considerably reduces anchorage capacity. These soils conditions promote a fast tree turnover (high mortality and recruitment) compared to terra-firme forests (*Korning & Balslev, 1994*; *Quesada et al., 2012*). As a result, smaller trees dominate forests on soil with physical restrictions and subjected to water saturation (*Schietti et al., 2016*), but frequent disturbances may also contribute to this pattern (*Toledo, Magnusson & Castilho, 2013*) by killing larger trees and promoting the restart of the successional process. In terra-firme, tree mortality is heterogeneously distributed across the landscape, with higher mortality rates found in valleys, where sandy soils with shallow water table are frequent and provide little support to tree anchorage (*Ferry et al., 2010*; *Toledo et al., 2011*; *Toledo, Magnusson & Castilho, 2013*; *De Souza, Dos Santos & Souza, 2017*). In particular, tree mortality by uprooting is more frequent during wetter periods (*Fontes, Chambers & Higuchi, 2018*) and on sandy soils with shallow water table (*Gale & Barfod, 1999*; *Toledo et al., 2012*), indicating that the substrate stability is a key factor affecting mortality, which in turn influences competition between trees.

Across Amazonia, neighborhood crowding (indicator of competition) negatively affected tree growth in forests with high wood density (*Rozendaal et al., 2020*), and the strength of competition increased with water availability (possibly due to higher basal area

of more humid forests) and declined with soil fertility, likely because fertile soils have higher density of low wood density species which seems to be less affected by competition. In the present study, tree density for terra-firme (516 stems ≥10 cm DBH ha$^{-1}$) and várzea (528 stem ha$^{-1}$) were similar, but palms represent 47% of all stems in várzea forest, whereas in terra-firme palms are very uncommon, thereby investment in stem height is probably smaller in várzea since competition for light may be less intense than in terra-firme. Indeed, competition for resources (light and nutrients) may play an important role, shaping the allometric relation between diameter and height in the beginning of tree ontogeny, driving fast growth in height, but the intensity may decrease since tall trees may fall down because of the lack of substrate support. The result is a steep slope for the height-diameter relationship restricted to smaller trees, which decrease as competition among taller trees become low.

Differences in the trade-off between lateral and apical growth, also can influence height-diameter relationships. In unstable soils, trees may allocate more resources to build large buttressed roots to resist against external forces and thus investing less in height. Trees with buttresses and stilt roots may be thick or thin, but trees with large trunk diameter have a higher probability of developing these support structures. Further, support structures were negatively related to the height: diameter ratio, showing that stout trees have a higher probability of developing support structures than slender trees (see *Alencar, de Castilho & Costa, 2023*). Also, support structures were more frequent in valleys where less stable soils are more common, indicating that trees develop such structures to stand on soils that do not provide stability (*Alencar, de Castilho & Costa, 2023*). Therefore, support structures seem to be often required for large trees with large crowns in unstable soil conditions, showing that differences in allocation of resources may affect height allometry between forest types.

## Variation in local model performance associated with forest type

We found that asymptotic models (Quadratic, Michaelis-Menten and Weibull) were the best fit for the data. The Weibull model showed a poorer fit in terra-firme and for all models in várzea and pooled data of both forests. These deviations can be associated with differences in non-linear relationships of height and diameter between forest types at both intra- (*Siliprandi et al., 2016*; *Nascimento et al., 2020*) and inter- (*Feldpausch et al., 2011*, *2012*; *Sullivan et al., 2018*; *Barbosa et al., 2019*) specific levels. Nonetheless, inter- and intra-specific variation in height between forest types cannot be accurately partitioned without controlling for tree age, which is correlated with height. Using a modified ANOVA approach (following *Lepš et al., 2011*), we estimated that the variation in height associated with species turnover between forest types was only 7.2%, while intra-specific variation was near zero (0.04%). Additionally, functional traits are related with several architectural characteristics of trees. For example, wood density is related to trunk diameter and volume, crown dimensions, wood elasticity and resistance to stem breakage (*King et al., 2006*, *Iida et al., 2012*; *Chave et al., 2009*), but there is no clear association with total height. However, intraspecific variation in the height-diameter relationship of *Goupia glabra*, a common canopy species in Amazonia, was associated with wood density, with taller trees showing on average denser wood (*Siliprandi et al., 2016*). In the present study, the correlation

between total height and wood density obtained with taxonomic information was low (Pearson correlation= 0.17 and 0.07 for terra-firme and várzea, respectively), but further investigation with field sampled wood density at species level is still necessary.

The asymptotic models chosen in this study described a relationship of very rapid increase in height relative to diameter for small- and medium-sized trees, but the rate of increment in height decreases very quickly for large trees (≥40 cm DBH). In other words, the expected height for large trees is lower than that expected for smaller ones. Most understory species and long-lived pioneer species did not reach estimated asymptotic heights in a southern Amazonian forest in Bolivia (*Poorter, Bongers & Bongers, 2006*), demonstrating that asymptotic models may not adequately fit to forests with abundant understory and pioneer trees. In this study, we demonstrated that incorporating species composition into the models improved precision by approximately 10% when using data from both forest types. This finding suggests that differences in model performance are in part influenced by species turnover across these forests.

High variation in height of large trees makes it difficult to find accurate models for height-diameter allometry since the number of these trees is often small in the datasets, and small datasets may result in inaccurate models (*Sullivan et al., 2018*; *Barbosa et al., 2019*). Nonetheless, we observed a higher number of large trees (≥90 cm DBH) in várzea (6) compared to terra-firme (three), but the lower model performance was detected for várzea. This suggests that a different numbers of large trees may be required for each forest type to achieve accuracy, possibly due to the wide variation in tree height y (*e.g.*, 7–37 in várzea *vs.* 35–43 m in terra-firme for trees ≥ 90 cm DBH). This may occurs because trees in várzea may allocate more resources to crown development (*Goodman, Phillips & Baker, 2014*) and structural stability (*e.g.*, buttressed roots) (*Alencar, de Castilho & Costa, 2023*) rather than height, a variation that diameter-based models may not fully capture. Environmental factors such as topography, soil texture and fertility, groundwater, flooding, and disturbances also influence large-tree traits (*e.g.*, growth, mortality), resulting in high variability in the height-diameter allometry within and between forest types (*Schietti et al., 2016*; *Fayolle et al., 2016*; *Siliprandi et al., 2016*; *Gorgens et al., 2021*). Additionally, the interaction between soil conditions and light competition may impact allometry. In Amazonia, shallow sandy soils prone to flooding lead to higher mortality (*Toledo et al., 2012*; *Fontes, Chambers & Higuchi, 2018*), favoring shorter trees with stability adaptations (*Alencar, de Castilho & Costa, 2023*). This high mortality creates an open canopy and a lower-competition environment for light, promoting lateral crown growth over height. In contrast, trees on stable, deep soils experience lower mortality and closed canopies (*Rozendaal et al., 2020*), which stimulates height growth to compete for light.

Height restriction for large trees may also be associated with hydraulic constraints. Drought affects disproportionately the growth and mortality of large tropical trees (*Nepstad et al., 2007*; *Bennett et al., 2015*) and this is attributed to vulnerability to hydraulic failure associated to large tree size (*Rowland et al., 2015*). However, tree diameter is often related to mortality and not height. Even though height is related to diameter, many studies (*Banin et al., 2012*; *Sullivan et al., 2018*; *Barbosa et al., 2019*) on tree height-diameter allometry of tropical forests (including this study for várzea), showed

limitations in height associated to large diameters, which can mitigate the exposure of crown to extreme dry conditions. The mechanism behind the height-dependent increase in mortality is not clear, since tall trees have several structural and functional adjustments to use and transport water more efficiently to mitigate the cavitation and carbon starvation during droughts, such as more efficient water usage and transport, as well as enhanced water uptake and storage capacity (*Fernández-de-Uña et al., 2023*). Also, recent findings from Amazonia basin wide inventories revealed a decreasing risk of mortality related to tree diameter and an increasing risk associated to species growth rate and dry season severity (*Esquivel-Muelbert et al., 2020*). Since the density of giant trees (>70 m in height) is associated with low wind disturbance and elevated light availability in the Northeastern Amazonia (*Gorgens et al., 2021*), further investigation about environmental stability and resource availability may be done for a better understanding of differences in height-diameter allometry between forest types in this region.

## Performance of local *vs* regional and pantropical height models

The local models outperformed the Guyana shield and pantropical models to predict tree height for terra-firme and várzea forest separately. General height equations for the tropics (*Feldpausch et al., 2012*) and even height models that include climatic indices, such as the pantropical equation by *Chave et al. (2014)*, may provide biased height estimates in forests dominated by shorter trees. Tree height in white-sand Campinarana forests of northern Amazonia was overestimated by 10–29% by the general Weibull model while underestimated by 8% by the pantropical model (*Barbosa et al., 2019*). However, at the regional scale, using pooled data from terra-firme and várzea forests, the pantropical model generated more accurate estimates of height. As climate variation between these sites may be responsible for part of the tree height variation, the index of environmental stress probably contributed to reduce bias. Nonetheless, site properties (*e.g.*, soil, topography, groundwater, flooding, disturbances, and species composition) may affect tree height causing interspecific and intraspecific variation (*da Silva et al., 2007*; *Siefert et al., 2015*; *Schietti et al., 2016*; *Fayolle et al., 2016*; *Siliprandi et al., 2016*) which cannot be predicted properly with continental scale models since it is difficult to incorporate all these site characteristics. For most remote sites, information is not available and satellite data may not provide enough resolution to describe the *in situ* variation.

Non-linear models that estimate maximum height (such as Michaelis-Menten, Weibull and Exponentials) were more precise in estimating height at local scale (see *Sullivan et al., 2018*, and *Barbosa et al., 2019*), whereas for pantropical or regional scale data, the simple power function (or derived forms: log-linear, log-log) showed the best performance (see *Feldpausch et al., 2011*, *2012*). Pantropical and regional datasets contain more large trees, which may exhibit a steeper slope within the large tree strata. Comparing the height-diameter allometry across continents, *Banin et al. (2012)* found steeper slopes for datasets (Asia and Africa) with a high number of tall trees (>40 m) compared to those datasets with few tall trees (South America and Australasia). The height estimates for large trees obtained using a power function were notably higher than those obtained using an asymptotic exponential function. Datasets with a limited number of tall trees are expected

to show high variation in height and possibly the asymptotic models are more suitable because of the estimate of extra parameters (such as maximum height) which bend the curve towards the center of the height distribution of large trees, resulting in a better error minimization compared to the power function.

*Chave et al. (2014)* recognized that some low-height forests may have height overestimated by the pantropical model and recommended the development of local equations, but our findings showed that the pantropical model failed to predict the height of high-height forests, underestimating by more than one fifth the height of trees in a terra-firme forest of the Guyana shield. The average of the climatic index (0.015) used in the pantropical model for the terra-firme forest fell within the range (−0.2–0.2) found for most tropical forests under water and temperature stress. Furthermore, the nutrient-poor and well-structured soils found in the terra-firme forest are similar to the predominant soils (Oxisols) found in the Amazon basin. Therefore, neither climate nor soils are expected to explain the unpredictability of the pantropical model for this terra-firme forest. However, *Gorgens et al. (2021)* showed that low wind disturbance and light availability are the main factors associated with high density of giant trees (>70 m in height) in Northeastern Amazonia. This evidence suggests that the climatic index in the pantropical equation is an inadequate surrogate for the specific climatic variables affecting tree height in the region. Other plausible hypothesis is that the sample size of the pantropical model was not large enough to capture most existing tree height variation in the Amazon basin. The same climatic or edaphic conditions may host very different vegetation structures and species composition at a mesoscale (up to 100 km$^2$) within the same forest type (see *Castilho et al., 2006*; *Damasco et al., 2013*), and such variation is not captured by pantropical height models. Indeed, compared to local models, pantropical or regional models tend to have a proportionately smaller sample sizes relative to the area they represent, potentially reducing their precision. However, field measurements are now facilitated with the use of portable technological devices, allowing fast and accurate (~1 m error) measurements of tree height (*Pereira et al., 2019*). Additionally, *Sullivan et al. (2018)* demonstrated that, at local scale, measurements of just 20 trees per forest can outperform the pantropical allometric model. Therefore, efforts to conduct such measurements must be made to provide enough data to allow the fitting of local-specific height-diameter models.

## Effect of species composition on diameter-height relationships

The inclusion of species identity, as well as controlling for phylogenetic autocorrelation in the height-diameter models, minimally improved the performance of local forest models. However, the inclusion of species identity contributed to increase the precision of models fitted to pooled data of both forests. The differences in species composition between terra-firme and várzea are well documented (*Campbell et al., 1986*; *Assis & Wittmann, 2011*; *Bredin et al., 2020*) and possibly captured part of the variation in height which was not explained by diameter alone. Nonetheless, intra-specific variation may increase the differences in allometry between forests (see *Fayolle et al., 2016*; *Siliprandi et al., 2016*). In the study region, *Carapa guianensis*, an abundant species shared between terra-firme and várzea (*Villacorta et al., 2023*), has shorter trees in várzea at the same size range

(nearly 25% and 40% shorter for trees with $10 \leq$ DBH $<30$ cm and $\geq 30$ cm DBH, respectively). However, in this study, only 1.2% of tree species (3.6% of individuals) were shared between terra-firme and várzea, demonstrating that inter-specific variation is more important than intra-specific variation to explain differences in height-diameter allometry. These differences in composition may generate variation in trait averages as show by the taller height (22.9 m) found for terra-firme compared to várzea (15.8 m). Nonetheless, it is difficult to separate the variation explained by interspecific variation in height from variation caused by environmental factors.

## Implications of tree height for biomass estimates

We showed that plot biomass calculated with height provided by local models developed in the present study was more accurate than biomass calculated using heights from regional and pantropical equations from *Feldpausch et al. (2012)* and *Chave et al. (2014)*, respectively. These findings show that the regional or pantropical scale models cannot represent most variation of tree height and the uncertainties can be propagated to plot-level biomass. Although the lack of local destructive biomass data results in uncertainties about the choice of the best biomass model for local forests, the use of height results in a decreasing in biomass in forest where trees are shorter than expected for a given diameter (*Nogueira et al., 2008a*) and contributes significantly to reduce the error in biomass estimates for forests dominated by shorter trees (*Feldpausch et al., 2012*, *Chave et al., 2014*). Trees of várzea were shorter than expected and thereby models without height (or even with heights from the Guyana shield and pantropical models) caused significant overestimation of the biomass. Conversely, trees of terra-firme have taller trees than expected for a given diameter and thus the underestimation of biomass in models that do not employ height or with height from the pantropical model. The use of local models would adjust downward (−13%) the estimates of biomass for várzea, and adjust upward (22%) the estimates for terra-firme. The adjustment of biomass has important economic implications for carbon-payment schemes under the Reducing Emissions from Deforestation and Degradation (REDD+) program. Estuarine white-water várzea forests have much smaller area (2.5 million hectares) compared to the terra-firme forests of the Guyana shield (148 million hectares; *Feldpausch et al., 2012*). Therefore, a 22% upward correction of tree biomass ($\geq 10$ cm DBH) for Guyana shield forests (average of 299 Mg ha$^{-1}$) will add 4.74 Pg to the carbon stocks while a 13% downward for white-water estuarine forests (average of 266.6 Mg ha$^{-1}$) will subtract only 0.04 Pg C, considering 48.5% of carbon in the dry biomass (*Nogueira et al., 2008b*). The net gain (4.7 Pg C) represents an important update for negotiations under the REDD+ payment schemes. Assuming the price of US $ 7.13 per Mg C (*Procton, 2024*), the potential gain for countries of the Guyana shield may be US $33.51 billion per year. Therefore, the modeling of tree height appears to be crucial for local forests, since general height-diameter relationships may vary substantially by forest type, which influences biomass estimation and consequently the REDD+ payment schemes.

Tree height calculated with local asymptotic models improved the accuracy of biomass estimates across size classes. However, higher uncertainties for biomass of large trees may

be due to their wide variation in height. *Feldpausch et al. (2011)* and *Feldpausch et al. (2012)* used asymptotic models (which estimate maximum height) to increase the precision of estimates of small and intermediate-sized trees of the Guyana shield because these trees represent most part of the biomass. Large trees (≥70 cm DBH) show a wide variation in height in this study, varying from 26 to 50 m (18 trees) in terra-firme and from 8 to 39 m (28 trees) in várzea. Therefore, only a concentrated effort to increase the sample size of these trees may improve the accuracy of height-diameter models. Increasing sample size of large trees across different forest types can increase the predictability of height models, as these trees show substantial variability in height within and between forest types. Consequently, using the estimates produced with these refined height models in biomass equations may reduce the error propagation in biomass estimates, as large trees contribute disproportionately to total forest biomass.

## CONCLUSIONS

Asymptotic (including Quadratic) height-diameter models fit better for trees from both terra-firme and várzea forests of Northeastern Amazonia. Such models outperformed the *Feldpausch et al. (2012)* and the *Chave et al. (2014)* height-diameter models when terra-firme and várzea forests were treated separately. Therefore, we highlighted the need for local models to increase the precision of height-diameter allometry for different forest physiognomies. Local models should be developed based on already available data and for new forest inventories conducted for scientific purposes or logging concessions. Published protocols (*Larjavaara & Muller-Landau, 2013*) can be used to measure tree height, and model selection can follow the approach of this study or those outlined by *Sullivan et al. (2018)*, allowing each forest site to have a specific equation for tree height.

Heights estimated from *Chave et al. (2014)* pantropical model and the biomass model with absence of height underestimated the biomass for terra-firme and overestimated for várzea forest since the former has taller trees and the latter shorter ones than expected for a given diameter. These uncertainties were mainly due to error for larger trees which have a wide variation of height. Therefore, modeling of tree height is needed for different forest types and more effort to sample large trees is needed to improve biomass estimates in Northeastern Amazonia.

### Funding

Coordination for the Improvement of Higher Education Personnel (CAPES) provided Aldine L. P. Baia a scholarship to undertake her Master's thesis. Financial support was provided by Federal University of Amapá (PAPESQ Program # 015/2015) and the National Council for Scientific and Technological Development—CNPq (Universal #409827/2021-5; #447432/2014-1; #459735/2014-4). The manuscript publication was supported by Foundation for the Support of Research in Amapá (#250.203.045/2019). José

Julio Toledo was supported by CNPq with a Research Productivity Scholarship (#316281/2021-2). The funders had no role in study design, data collection and analysis, decision to publish, or preparation of the manuscript.

## Grant Disclosures

The following grant information was disclosed by the authors:
Coordination for the Improvement of Higher Education Personnel (CAPES).
Federal University of Amapá (PAPESQ Program): # 015/2015.
National Council for Scientific and Technological Development—CNPq (Universal): #409827/2021-5, #447432/2014-1 and #459735/2014-4.
Foundation for the Support of Research in Amapá: #250.203.045/2019.
CNPq with a Research Productivity Scholarship: #316281/2021-2.

## Competing Interests

The authors declare that they have no competing interests.

## Author Contributions

- Aldine Luiza Pereira Baia performed the experiments, analyzed the data, prepared figures and/or tables, authored or reviewed drafts of the article, and approved the final draft.
- Henrique E. M. Nascimento conceived and designed the experiments, analyzed the data, prepared figures and/or tables, authored or reviewed drafts of the article, and approved the final draft.
- Marcelino Guedes conceived and designed the experiments, authored or reviewed drafts of the article, and approved the final draft.
- Renato Hilário performed the experiments, authored or reviewed drafts of the article, and approved the final draft.
- José Julio Toledo conceived and designed the experiments, performed the experiments, analyzed the data, prepared figures and/or tables, authored or reviewed drafts of the article, and approved the final draft.

## Field Study Permissions

The following information was supplied relating to field study approvals (*i.e.*, approving body and any reference numbers):

The permission for botanical sampling was granted by the Instituto Chico Mendes de Conservação da Biodiversidade—ICMBio.

## Data Availability

The raw data is available in the Supplemental File.

## Supplemental Information

Supplemental information for this article can be found online at http://dx.doi.org/10.7717/peerj.18974#supplemental-information.

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
