# Peer review of "Tree height-diameter allometry and implications for biomass estimates in Northeastern Amazonian forests"

_PeerJ, doi:10.7717/peerj.18974_

## Round 0.1 · original submission · Major Revisions

Dear authors,

I have solicited two reviews of your manuscript entitled “Tree height-diameter allometry and implications for biomass estimates in northeastern Amazonian forests”.

The reviewers agreed that the topic of tree allometries in the Amazonian Forest is interesting to analyse forest development. However, the interest of this study is not clearly addressed in the introduction.

The reviewers also stated that the manuscript should be improved for clarity of the methods, particularly the models used to explore the allometries, and the display of the results in the figures. They also claim that the introduction and the discussion should be more focused on ecological questions based on the previous literature.

Lastly, they highlighted that some sentences are not understandable and then, the English needs to be reviewed by a native speaker.

Based on the reviewers’ and mine own concerns, I think your manuscript needs improvements. I invite you to respond to each point raised by myself and the reviewers and submit a revised manuscript.

My specific comments are detailed below:

Mayor concerns

1) The ecological question that the study is trying to respond is not clearly described and it should be clearly mentioned in the abstract and in the introduction.

2) Only one forest (with five sites in 5 km side area) for each forest type (várzea and terra-firme) were investigated and then tree replicates at each site might be thus considered as pseudoreplicates with respect to statistical tests for site effects. Hence, due to the absence of real replicates for each of the investigated habitats, any significant differences between the two sites might or might not be related to site-specific edaphic or climatic conditions. Therefore, the authors should be cautious in order to discuss site differences. The only way to overcome this issue is sampling more sites and at different forests for each forest type to get real replicates.

3) Where are the results of random errors indicated (Line 263)? The relationships the authors explore, should be modelled using a Generalized Mixed Model. Note that random effect should include species as source of variation. Please, check also this method to get the accuracy of your models, Nakagawa, S, Schielzeth, H (2013) A general and simple method for obtaining R2 from generalized linear mixed-effects models. Methods in Ecology and Evolution, 4,133–142.

4) Moreover, the authors analysed several species, then the results may be biased because the species with available data are not randomly distributed across the whole diversity of trees. To account for this potential bias, the statistical analyses should be controlled by the phylogenetic relationships of the analysed tree species because closely related species can share the same allometric relationships because of their common evolutionary history.

5) Lines 328-340. Regarding this rationale, it should be better to classify the measured trees into tree types (for example, monocotiledons, dycotiledons, gymnosperms) in order to better understand the biological limits related to anatomical and physiological features. Remember also that tree height is restrained by water conduction and it should be discussed. Please, see Bennett AC, McDowell NG, Allen CD, Anderson-Teixeira KJ (2015) Larger trees suffer most during drought in forests worldwide. Nature Plants, 1, 15139.

6) Also, the inter and intra-interspecific differences can be checked. The amount of variation explained by each level can be computed using hierarchical partitioning.


Minor concerns

7) Line 52. What do you mean by “wood specific gravity?” Should be much clear using wood density.

8) Line 66. What do you mean by “tree stature”? Should be much clear using tree height.

9) Line 71. What do you mean by “it varies by almost twelve times between forests”?

10) Line 73. Are you sure about it? Now there are several climatic databases with high resolution data.

11) Line 120. What do you mean by “forest physiognomies”? Should be much clear using forest type.

12) Lines 144-147. Which DBH categories do you have? Please, explain it better.

13) Lines 154-L158. How many species did you analyse? Please, add a list with the sample size for each species in supplementary data.

14) Lines 247-L253 These results need statistical analyses testing the differences between sites and forest types.

17) Figure 2. Grey scale legends are difficult to follow. Please, clarify it.

18) Are the percentages of Figure 2 different among the models? Please, test it.

Sincerely,
Lucía DeSoto

Reviewer 1 ·

Basic reporting

The manuscript is of good quality with some unclear sentences which I point out later.

Experimental design

The research questions are defined well enough. Methods have been well described. The study is not based on experiments.

Validity of the findings

The validity of the findings is good.

Additional comments

The manuscript of high quality – much higher than manuscripts that I peer-review on average. However, it was not of high interest to me. I would had been more interested in ecological or biomechanical discussion based on the results. There was a little bit of such discussion but I disagree with some of the statements. The reasoning around line 318 that wetland trees are shorter for a given diameter because the soil is less stable is not satisfactory. Trees could have evolved to have larger buttresses, root collars or roots in labile soil but still have thin trunks. If trees in both habitats have an equal risk of falling in a storm (maybe not true but roughly true probably), then larger heights for a given diameter could result from 1. lower storm wind speeds in that region, 2. from lower wind speeds due to protection from neighbouring trees, 3. from smaller crowns or 4. from stronger wood. Based on wood densities 4 seems not to be the case. Also 1 is unlikely. Could it be that the palms in the wetland lead to larger crowns and also stronger winds in non-palm crowns if the palms are shorter. Another piece of critique is that the logic that the dryland trees compete more for light causes them to be taller for a given diameter does not sound reasonable to me. Trees adjust their dimensions based on swaying in wind and simply increasing height for light competition would lead to a likely fall. More severe light competition would force trees to have both larger diameters and heights if risks are not increased.

Line 30
Instead of “early published pantropical height models”, maybe: pantropical height models published earlier.

40
The sentence needs restructuring. Also the next one. Maybe: … error in height estimation of large trees …

43
Instead of “must be”, better: is.

49
This sentence is somewhat unclear. Exactly what is causing the uncertainty?

52
If the references say that these decrease errors then instead of “may increase”, better: increase.

67
Shorter for a given diameter or shorter in general.

71
Exactly what varies 12 times? I guess there can also be in increase (not a reduction).

74
Again unsure if this is about height or height for a given diameter (probably later similar uncertainties so please check the whole manuscript).

77
Better: … where local biomass …

80
Again you are referring to height variation for a given diameter. Not sure how to write this better. Maybe in the beginning you e.g. have: height for a given diameter (refers hereafter to height for a given diameter).

86
I think that “downward” is not easily understandable here.

102
Better without “unique”.

106
I don’t like these words. What would be a climate that is not limiting height? Would tree height be then infinite?

109
Instead of “both” better: These.

122
“left” vague.

127
You can remove “and a dry season from August to November”.

137
Probably: follows.

331
I am sorry but do not understand the logic around here. Asymptotic curves could be ok for short trees if their diameter is never large.

333
Instead of “size” better to have: diameter.

Reviewer 2 ·

Basic reporting

no comment

Experimental design

The article deals with a very relevant topic, especially when we talk about allometry to forest species present in the floodplain of the Amazon forest. Something little investigated in forest science. However, there are some questions that were not well answered in the manuscript that could compromise the use of the proposed models:

1º - Why were biological growth models used in their natural form? in the definition of Huxley and Teissier (1936) allometric models aim at the relationship between production rates of different parts of a living organism. These models with asymptotic parameters, in this configuration, are not suitable for this type of modeling and the authors cite the work of Feldpausch et al. (2012) who affirm this.

2º The statistical assumption for using nonlinear biological growth models to adjust the hypsometric relationship has already been tested by several works, but none, including your work, have not tested the biological efficiency of these models. Those who work with forest plantations, whose forest age is known, do not apply these models to old-age forests, since the asymptotes of these models are statistically equal to the average height and the other parameters of the models have little weight in the adjustment of the model (Huang et al. 1992; Zhang, 1997; Machado et al 2008; Sharma et al 2016). The same recommendation is made when there is a high variability in diameter for the same height, as happens in forests with different ages and species (Nascimento et al. 2020 - https://doi.org/10.1016/j.tfp.2020.100028). I suggest including a biological trend test of the best performing models, such as the Theil and Graybill test, to support your results by verifying whether these models present biological realism of the hypsometric relationship.

3º All graphical analysis need to be improved. The Figure 2 and 3 are not cristal clear what the Authors want to present to the readers. I understood that it is a "kind" of error decompositions, but i would rather see total and systematic error side by side for each variable in the same frame/panel. May be easier to demostrate the error tendency using a residuals or observed versus estimaded graphs.

Validity of the findings

no comment

---

## Round 0.2 · Major Revisions

Please respond to these detailed reviews in an appropriate revision

Reviewer 3 ·

Basic reporting

The introduction situates the study within the broad field of tropical forest ecology, particularly in the context of biomass estimation, which is crucial for understanding global carbon cycles.

While the introduction includes a comprehensive review of relevant literature, it could be more concise and focused. The extensive details on species composition, soil characteristics, and climatic factors, although relevant, dilute the primary research focus.

The repeated citation of some authors and studies could be optimized to prevent redundancy (for example, Chavé et al. (2014) is cited seven times in the introduction). Streamlining these references would enhance the conciseness of the introduction, thereby increasing its effectiveness in communicating the significance and objectives of the study.

Here are some examples where the language in the introduction could be modified for clarity:
Lines 49-51: "Tree diameter is the most common variable used to estimate biomass indirectly, but the insertion of other traits such as tree height, crown size and wood density increase the precision and accuracy."
Suggested: "Tree diameter is commonly used to estimate biomass, but incorporating traits such as height, crown size, and wood density improves precision and accuracy."

Lines 55-58: "The inclusion of height in allometric models (Chave et al., 2005; Chave et al., 2014; Feldpausch et al., 2012) and the higher variation in tree height along climatic and edaphic gradients bring to light more uncertainties about forest biomass estimates..."
Suggested: "Including height in allometric models (Chave et al., 2005; Feldpausch et al., 2012; Chave et al., 2014) and considering variation in tree height along climatic and edaphic gradients introduces further uncertainties in biomass estimates..."

Lines 88-93: "Species composition varies widely at local (Damasco et al., 2013; Toledo et al., 2017) and regional (ter Steege et al., 2006) scales in Amazonia, which may partially explain the variation in height-diameter relationships. Nonetheless, how much variation in height-diameter is accounted for variation in species composition is subject to investigation between Amazonian Forest types."
Suggested: "Species composition varies widely across local (Damasco et al., 2013; Toledo et al., 2017) and regional (ter Steege et al., 2006) scales in Amazonia, partially influencing height-diameter relationships. Nonetheless, the extent to which species composition affects these relationships remains under investigation."
Please do so throughout the manuscript.

Experimental design

Methodology is clear, and sufficiently detailed; the methods used, calculations performed and statistical tests support the results obtained. However, I have some observations.
₋ Although it can be inferred that the study focuses on the height-diameter relationship and biomass estimation in different forest types, the research question is not clearly stated in the methodology section. This makes it harder to grasp the specific purpose of the study.
₋ Species identity was included as a random factor (line 234) in the height-diameter models, but there is insufficient explanation of how the differences in species that were not fully identified (e.g., trees identified only to genus or family level) might affect the model’s precision. Please explain.
₋ The methodology mentions that plots are separated by at least 1 km (lines 186-187), but it does not explain the rationale behind this distance selection. If these plots were chosen based on specific environmental or geographic criteria, this should be clarified to avoid the perception of bias in the study site selection.
₋ On the other hand, a weakness of the study is the sample size. The authors used 5 sites of 0.5 ha separated 1 km from each other in the terra-firme forest type, which covers an area of more than 4.5 million ha, which is definitely a sample that does not capture all the variability in height. How do the authors justify the validity and applicability of the local models developed with so little data for such a large area?

Validity of the findings

₋ The results are based on quantitative data and statistical analysis, which minimizes subjectivity. However, the interpretation of model performances and error types could introduce some level of subjective judgment. For example, the assessment of model fits and the discussion on random versus systematic errors (lines 357-364) involve interpretative decisions. Ensuring that these interpretations are based on robust statistical evidence and clearly defined criteria.
₋ The results mention that models show significant deviations from expected relationships for large trees (lines 471-473). Please fully explain the reasons behind these findings.
₋ It is mentioned that asymptotic models do not adequately describe the height of large trees (lines 636-637); please mention the possible biological reasons of this limitation of the models.
₋ Although the need to model tree height for different forest types and to make additional effort to sample large trees is mentioned, it must be discussed in detail how these actions might reduce uncertainties or improve biomass estimates.
₋ The conclusion highlights the need for local models to increase the accuracy of height-diameter allometry in different forest types, which is an important finding. What concrete steps should be taken to apply these local models?
₋ The authors mention in the discussion (lines 605-609) that models using height generate better results than those using only diameter or than the pantropical model. They also point out that some models overestimate and others underestimate aboveground biomass (AGB). Against which AGB values are the estimated AGB with these models being compared? Which AGB is considered ‘real’ or ‘observed’? To compare the biomass estimates in the study, AGB values obtained from destructive sampling, where sample trees have been felled and weighed green and dry (in fact, the authors mention this in the discussion, lines 611-615), are needed. Include this in the methodology and emphasise it in the discussion.

Additional comments

Formalities and specific observations (line number refers to the pdf manuscript):

Lines 20-21: Uncertainty in tree height estimates does not affect aboveground biomass, it affects biomass estimates, as these estimates have implicit height error, which is cumulative. Please rewrite.

Throughout the manuscript aboveground and above-ground biomass are used, please use only one form.

Lines 188-191: Clarify the criteria for stratification of the trees in the sample, as saying “trees with BHD≥ 1 cm” introduces confusion; for example, a BHD of 20 cm is greater than 1 cm. I recommend clarifying this part of the methodology as follows: trees with DBH between 1 cm and 10 cm, trees larger than 10 cm and smaller than 30 cm, or something like that.

Line 228: What is the pantropical climate model? Eqn. 3 or Eqn. 4? Indicate in brackets the equation number for each model. Do so throughout the manuscript.

Lines 567-569: The following sentence is confusing: “General equations for the tropics (Feldpausch et al., 2012) and even models which include climatic index (Chave et al., 2014) pantropical model may provide biased estimates of height in forests dominated by shorter trees…”. I think that one or more words are missing before “pantropical”.

Reviewer 4 ·

Basic reporting

Tree height-diameter allometry of trees is worthwhile to be investigated, in order to improve the estimation of biomass and carbon inventory using allometric equations. This study presents extended amount of tree allometry data from two Northeastern Amazonian forests with contrasting conditions and applies advanced analysis tools such as phylogenetic autocorrelation and error decomposition. This manuscript seems interesting, valuable, but difficult to read unfortunately. Even though I found authors did lots of efforts to revise the manuscript in accordance with the comments from the editor and reviewers in the first round, I still found several points that make the manuscript difficult.

- There are two tables and three figures in the main text. Those materials have too many information and seem complicated; therefore, I failed to get the key findings of this study from the materials in the main text. I suggest these materials can be more focused and simplified to improve the readability. In addition, more materials are included the supplementary. Those supplementary materials are referred in the results section too often, resulting in disturbing the flow of reading. Some of them can be included in the main text with an improved format.

- A discussion section seems a bit long and redundant. I partially understand that the too narrative discussion in this version is resulted from the rebuttal process in the previous round. Please reorganize the discussion section and make subsections. I would like to read the discussion research questions by research question.

Experimental design

- As far as I understand from reading this manuscript, the key finding of this study is a local allometric model is more accurate than regional or generalized ones. While the data, analyses, and background of this study seems attractive, the key findings and implications don’t. The difference of tree height-diameter allometry between two regions is just explained something of regional difference. Phylogenetic structure is minor contributor to explain these differences against the authors’ expectations. I wished to see more novel and surprising findings; but, I think that’s the best the authors can do with the current research design and satisfy the publish-ability. Here, the pseudo-replicate issue, which was raised in the first round, should be re-considered again. I don’t say more replicates should be made in this moment. Rather, my point is that the value and limitation of current research design can be briefly stated with the concerns of pseudo-replicate in a relevant place.

Validity of the findings

- Several advanced analytic tools that I am unfamiliar with are used to test the research question. For example, I am wondering how the species identity was included as a random factor in the GLMM. Statistical formula may be helpful to understand the principle and practice of those statistical models and/or tests.

- How was the accuracy of allometric models in biomass estimation tested? I don’t ask the statistics. Are there observed biomass values from harvest, representing “true values?

Reviewer 5 ·

Basic reporting

See attached review.

Experimental design

See attached review.

Validity of the findings

See attached review.

Additional comments

See attached review.

Annotated reviews are not available for download in order to protect the identity of reviewers who chose to remain anonymous.

---

## Round 0.3 · Major Revisions

Dear authors, I ask you to take into account the reviewers' comments and send the final version of your manuscript for publication.

Reviewer 1 ·

Basic reporting

-

Experimental design

-

Validity of the findings

-

Additional comments

I peer-reviewed this manuscript already in 2021. I checked that my critique was properly addressed already in August this year. Therefore, I do not see the need review the manuscript again.

Reviewer 4 ·

Basic reporting

I can't imagine the amount of effort the authors put into revising the manuscript. A great deal of explanation and discussion has been added. As a result, the manuscript is now more clear and comprehensive, even though it reads quite lengthy. I don't recommend adding anything more. Instead, I strongly suggest that the topic dealing with the biomass estimates comparison be removed or minimized. As mentioned by other reviewers in previous reviews, we don't know the true observed biomass values, although harvesting to get observed values is infeasible.

Therefore, I believe that the biomass estimates calculated using measured height, estimated height, or without height cannot be definitively determined as over or under-estimated without observed values. How can the biomass allometric equations be guaranteed to accurately estimate the biomass of the studied trees? Since the pantropical height-diameter models failed to accurately estimate height at the local level according to this study, the pantropical biomass allometric models which was adopted for estimate biomass would likely fail too. Therefore, while I agree that the local height-diameter models suggested by this study are useful for biomass estimates in northeastern Amazonian forests, I don't agree that they improve the accuracy of biomass estimates.

Experimental design

no comment

Validity of the findings

no comment

Additional comments

Below are some very minor details.

L. 158 Include the thesis paper in the reference list and cite it in the text, instead of the URL link in the text
L. 172-173 The symbols for minute and second of the coordinate are slightly different between longitude and latitude (' ' " ”). Check them.

·

Basic reporting

The article addresses an important ecological issue that aims to put into practice the understanding of the allometric dependence of tree height on their diameter. This question is addressed in one of the world's most diverse forests, which is of interest to a wide range of researchers and gives hope that the results can be extended to other environmental conditions around the world

Experimental design

The experimental design is carefully presented in the article, which makes the information provided repeatable. The experimental design is correct, so the conclusions are scientifically sound

Validity of the findings

The aims stated at the beginning of the article have been logically and reasonably achieved. The results are of practical importance and significantly extend our understanding of the aleometric dependencies between tree properties.

Additional comments

1. It is better to avoid references to literary sources in the abstract.
2. The abstract should not detail the methods by listing the stages of the study.
3. DBH - should be deciphered at the first mention; in the article, the decoding is given only for the 5th mention of DBH.
4. Figure 1 allows to ask the question: why did the authors not consider the simplest relationship - a linear one to describe the dependence? Also, to assess the validity of the regression model, it is appropriate to consider the distribution of the model residuals - they should follow a normal distribution law. It is obvious from Figure 1 that for no model will the residuals follow a normal distribution law.
5. The information presented in Figure 2 does not provide additional information and is of a technical nature. If the authors consider it necessary to provide relevant figures, they would be more appropriately presented in the Appendix.

---

## Round 0.4 · accepted · Accept

Dear authors, I congratulate you on the acceptance of this article for publication. I hope that you will continue your research and send your manuscripts to our journal again

Reviewer 4 ·

Basic reporting

no comment

Experimental design

no comment

Validity of the findings

no comment

Additional comments

no comment

·

Basic reporting

All recommendations have been taken into account. Recommended for publication

Experimental design

All recommendations have been taken into account. Recommended for publication

Validity of the findings

All recommendations have been taken into account. Recommended for publication

Additional comments

All recommendations have been taken into account. Recommended for publication